

# Trends in polar ozone loss since 1989: First signs of recovery in Arctic ozone column

Andrea Pazmiño[1], Florence Goutail[1], Sophie Godin-Beekmann[1], Alain Hauchecorne[1], Jean-Pierre Pommereau[1], Martyn P. Chipperfield[2,3], Wuhu Feng[2,4], Franck Lefèvre[1], Audrey Lecouffe[1], Michel Van Roozendael[5], Nis Jepsen[6], Georg Hansen[7], Rigel Kivi[8], Kimberly Strong[9], Kaley A. Walker[9] and Steve Colwell[10]

[1]LATMOS/IPSL, UVSQ, Université Paris-Saclay, Sorbonne Université, CNRS, Guyancourt, France
[2]School of Earth and Environment, University of Leeds, Leeds, UK
[3]National Centre for Earth Observation, University of Leeds, Leeds, UK
[4]National Centre for Atmospheric Science, University of Leeds, Leeds, UK
[5]Belgian Institute for Space Aeronomy (BIRA), Brussels, Belgium
[6]Danish Meteorological Institute, Copenhagen, Denmark
[7]Norwegian Institute for Air Research, Kjeller, Norway
[8]Finnish Meteorological Institute, Sodankylä, Finland
[9]Department of Physics, University of Toronto, Toronto, Canada
[10]British Antarctic Survey, Cambridge, UK

*Correspondence to*: Andrea Pazmiño (andrea.pazmino@latmos.ipsl.fr)

**Abstract.** Ozone depletion over the polar regions is monitored each year by satellite and ground-based instruments. In this study, the vortex-averaged ozone loss over the last three decades is evaluated for both polar regions using the passive ozone tracer of the chemical transport model TOMCAT/SLIMCAT and total ozone observations from Système d'Analyse par Observation Zénithale (SAOZ) ground-based instruments and Multi-Sensor Reanalysis (MSR2). The passive tracer method allows us to determine the evolution of the daily rate of column ozone destruction, and the magnitude of the cumulative loss at the end of the winter. Three metrics are used to estimate the linear trend since 2000 and to assess the current situation of ozone recovery over both polar regions: 1) The maximum ozone loss at the end of the winter; 2) the onset day of ozone loss at a specific threshold and 3) the ozone loss residuals computed from the differences between annual ozone loss and ozone loss values regressed with respect to sunlit volume of polar stratospheric clouds (VPSC). This latter metric is based on linear and parabolic regressions for ozone loss in the Northern and Southern Hemispheres, respectively. In the Antarctic, metrics 1, and 3, yield trends of -2.3 and -1.8% dec$^{-1}$ for the 2000-2021 period, significant at 1 and 2 standard error (σ), respectively. For metric 2, various thresholds were considered, all of them showing a time delay for when they are reached. The trends are significant at the 2σ level and vary from 3.5 to 4.2 day dec$^{-1}$ between the various thresholds. In the Arctic, metric 1 exhibits large interannual variability and no significant trend is detected; this result is highly influenced by the record ozone losses in 2011 and 2020. Metric 2 is not applied in the Northern Hemisphere due to the difficulty of finding a threshold value in a consistent number of winters. Metric 3 shows a negative trend in Arctic ozone loss residuals of -1.7 ±1% dec$^{-1}$, significant at 1σ level. This is therefore the first quantitative detection of ozone recovery in the Arctic springtime lower stratosphere.



## 1 Introduction

The first signs of healing of the ozone layer in the polar regions linked to the decrease of ozone-depleting substances (ODSs) was detected in Antarctica by Yang et al. (2008), who showed a statistically significant levelling off of the decrease in total ozone during Spring and by Solomon et al. (2016), who presented evidence of a statistically significant increase of total ozone in the depletion period. This increase was confirmed by later studies using measurements (e.g., de Laat et al, 2017; Kuttippurath et al., 2018; Pazmiño et al., 2018; Weber et al., 2018, 2021) and model simulations (e.g., Strahan et al., 2019). In contrast, in the Arctic, the large variability in meteorological conditions prevents detection of ozone recovery as shown by the recent trend study of Weber et al. (2021). Chemistry-climate models (CCMs) predict that climate change due to increasing greenhouse gases (GHGs) will accelerate ozone recovery in the Arctic due to the possible enhancement of the Brewer-Dobson circulation (BDC) (WMO, 2018). An early return of ozone to 1980 levels by 2034 is predicted by models used in the Chemistry-Climate Model Initiative (CCMI)-1 project (Dhomse et al., 2018). In the last Ozone Assessment Report (WMO, 2022) new analyses considering a small set of CMIP6 (Coupled Model Intercomparison Project Phase 6) models show that Antarctic ozone recovery to pre-depletion (1980) levels is sensitive to different climate change scenarios, while Arctic ozone recovery is about 11 years later for some scenarios compared to the projections in the 2018 Ozone Assessment Report (Chipperfield, Santee et al., 2023).

On the other hand, by analysing four reanalysis datasets, von der Gathen et al. (2021) find that Arctic winters are becoming colder and suggest that some GHG scenarios might favour the occurrence of large ozone depletion events. Polvani et al. (2019) show by a multi-model analysis that 60% of the modelled BDC trends over the 1980-2000 period could be attributed to ODSs. The authors also projected a strong deceleration of the BDC for the 2000-2080 period due to the decrease of ODS concentrations, counteracting the effect of increasing GHGs. However, the expected decline of ODSs after the full phase-out of production/consumption of chlorofluorocarbons (CFCs), halons, and carbon tetrachloride in 2010 under the Montreal Protocol has been questioned following the work of Montzka et al. (2018). They discovered an enhancement of CFC-11 emissions after 2012 that continued increasing during the 2014-2017 period. In addition to the illicit production of "controlled" ODSs, increasing emissions of non-controlled chlorinated very short-lived substances (VSLSs) have been observed (e.g., Claxton et al., 2020) adding a significant amount of ozone-depleting chlorine to the atmosphere (Chipperfield et al., 2020).

Continued observations of ozone on-board different platforms (ground-based, balloons, aircraft and satellites) in synergy with model simulation are necessary to assess the recovery of the ozone layer in the context of climate change and uncontrolled or illicit emissions that can impact ozone evolution. Episodic natural events such as volcanic eruptions can also interfere with the detection of ozone recovery (WMO, 2022 and reference within). More recently, wildfires events impacting stratospheric aerosol loading coincided with large ozone depletion in both polar regions. In the Arctic, the enhancement of stratospheric aerosols by Siberian fires in mid-2019 (Ohneiser et al., 2021), which remained in the polar region for a year, could have impacted the 2019-2020 Arctic winter that was characterized by a record ozone depletion (e.g., Manney et al., 2020; Bognar et al., 2021). In the Antarctic, the Australian Black Summer wildfires in 2019/2020 season (Khaykin et al., 2020; Peterson et



al., 2021; Tencé et al., 2022; Solomon et al., 2023) could have also influenced the large and long-lasting depletion during the 2020 Southern Hemisphere winter/spring.

For the detection of ozone recovery in Antarctica, different metrics have been used such as vortex area, minimum or average ozone at different months, occurrence of loss saturation and ozone mass deficit at different thresholds. During the last two decades, large variability has been observed in the area inside the vortex over which ozone concentrations/columns(?) are below various thresholds (Pazmiño et al., 2018). In the Arctic, two strong ozone depletions have been observed in the last two decades leading to very low ozone values in March and April 2011 (e.g. Manney et al., 2011; Pommereau et al., 2013) and

March 2020 (e.g. Manney et al., 2020; Wohltmann et al., 2020; Bognar et al., 2021; Feng et al., 2021; Wohltmann et al., 2021). The purpose of this study is to evaluate the long-term variability of ozone and separate the effect of chemical and dynamical processes in both polar regions in the context of current ODS and GHG evolutions by using a synergy between measurements and model simulations. The amplitude of ozone depletion has been monitored every year since the beginning of 1990s by comparison between total ozone measurements by Système d'Analyse par Observation Zenithale (SAOZ) UV-Vis

spectrometers (Pommereau and Goutail, 1988a) deployed in Antarctica and in the Arctic combined with multi-sensor reanalysis (MSR2) datasets (van der A et al., 2010), and the simulated "passive" ozone column by the TOMCAT/SLIMCAT 3-D chemical transport model (CTM) (Chipperfield, 2006; Feng et al., 2021) in which ozone is considered as a passive tracer (e.g. Feng et al., 2005). The method allows us to determine the evolution of the daily rate of total ozone depletion and the amplitude of the cumulative loss at the end of the winter.

This paper is organized as follows. Section 2 presents ozone datasets from the SAOZ instrument and MSR2. Section 3 describes the method used to calculate ozone loss inside the vortex. The analyses of recent winters in both polar regions are presented in Section 4. Section 5 introduces the ozone trend analysis. Conclusions are presented in Section 6.

## 2 Data

In order to estimate ozone depletion in the polar regions, ground-based SAOZ ozone columns and ozone MSR2 data reanalysis

as well as the modelled TOMCAT/SLIMCAT ozone are used.

### 2.1 SAOZ ground-based instrument

The SAOZ (Pommereau and Goutail, 1988a) instrument is part of the international Network for the Detection of Atmospheric Composition Change (NDACC, De Mazière et al., 2018) and French Aerosols, Clouds and Trace Gases Research Infrastructure (ACTRIS). The data used in this work are those of eight SAOZ stations distributed around the Arctic and three around

Antarctica (Table 1). SAOZ is a passive remote-sensing instrument that measures sunlight scattered from the zenith sky allowing precise measurements of stratospheric constituents during twilight (sunrise and sunset) for solar zenith angles (SZA) between 86 and 91°. It allows measurements throughout the winter season at latitudes near the polar circle. The retrieval method used by SAOZ is Differential Optical Absorption Spectroscopy (DOAS) (Solomon et al., 1987; Pommereau and



Goutail, 1988a, 1988b; Platt and Stutz, 2008) which is suitable for the detection of minor gases in the atmosphere. The measured slant columns of ozone and $NO_2$ are retrieved twice a day and converted to vertical columns using air mass factors (AMF) calculated by means of the UVSPEC/DISORT radiative transfer model (Mayer and Kylling, 2005). The SAOZ V2 retrieval applied in this work uses a multi-entry database of TOMS version 8 (TV8) ozone and temperature profile climatology (McPeters et al., 2007). Ozone is measured in the visible Chappuis bands (450-550 nm) where cross sections are weakly dependent on temperature, and $NO_2$ is measured in the wavelength range 410-530 nm using low-temperature cross sections (220 K). Spectral analysis and AMF settings follow the recommendations of the NDACC UV-Vis Working Group (Hendrick et al., 2011). The ozone and $NO_2$ vertical columns used here are sunrise and sunset means. Total ozone is retrieved with a precision of 4.5% and a total accuracy of 5.9% while $NO_2$ morning and evening columns are obtained with 10–15% accuracy (Pommereau et al., 2013).

**Table 1. Arctic and Antarctic stations included in the study: latitude, longitude and measurement periods of SAOZ datasets and the MSR2 assimilated data set.**

| Station | Lat, Lon | SAOZ dataset period | MSR2 dataset period |
|---|---|---|---|
| Eureka, Nunavut | 80.1°N, 86.4°W | 2005-2020 | 1990-2022 |
| Ny-Alesund, Svalbard | 78.9°N, 11.9° E | 1991-2022 | 1990-2022 |
| Thule, Greenland | 76.5°N, 68.8°W | 1999-2003, 2005-2016 | 1990-2022 |
| Scoresbysund, Greenland | 70.5°N, 22.0°W | 1991-2017, 2019-2022 | 1990-2022 |
| Sodankyla, Finland | 67.4°N, 26.6° E | 1991-2022 | 1990-2022 |
| Sondre Stromfjord, Greenland | 67.0°N, 50.6°W | 2018-2022 | 1990-2022 |
| Zhigansk, Russia | 66.8°N, 123.4° E | 1992-2013 | 1990-2022 |
| Salekhard, Russia | 66.5°N, 66.7°E | 2002-2016 | 1990-2022 |
| Marambio, Antarctica | 64.2°S, 56.7°W | - | 1989-2021 |
| Dumont d'Urville, Antarctica | 66.7°S, 140.0°E | 1989-2021 | 1989-2021 |
| Rothera, Antarctica | 67.6°S, 68.1°W | 2007-2021 | 1989-2021 |
| Syowa, Antarctica | 69.0°S, 39.6°E | - | 1989-2021 |
| Neumayer, Antarctica | 70.7°S, 8.3°W | - | 1989-2021 |
| Terra Nova, Antarctica | 74.8°S, 164.5°E | - | 1989-2021 |
| Concordia, Antarctica | 75.1°S, 123.4°E | 2007-2021 | 1989-2021 |
| Halley, Antarctica | 75.6°S, 26.8°W | - | 1989-2021 |

The difference between sunset and sunrise $NO_2$ total columns is calculated at each SAOZ station to follow the amplitude of the $NO_2$ diurnal cycle and to assess whether denitrification occurred inside the vortex that could promote ozone loss. SAOZ data are available on the NDACC database (https://www-air.larc.nasa.gov/missions/ndacc/) and the SAOZ webpage (http://saoz.obs.uvsq.fr/).

**2.2 Multi-Sensor Reanalysis (MSR2)**

In this study, daily SAOZ ozone data corresponding to the mean sunrise-sunset value are merged with daily MSR2 ozone columns. The MSR2 ozone dataset comprises daily assimilated gridded ozone columns at 12:00 UT at a spatial resolution of



0.5° × 0.5° in both hemispheres. The TM3-DAM CTM (simplified version of TM5, Krol et al., 2005) is used to assimilate 14 polar-orbiting satellite datasets, already corrected for SZA dependency, stratospheric temperature, and other parameters by comparisons with ground-based datasets from Dobson and Brewer networks which are part of the World Ozone and Ultraviolet

Data Center (WOUDC) (see van der A et al., 2010, 2015 for a detailed description). The data covering the 1989-2022 period are available from the Tropospheric Emission Monitoring Internet Service (TEMIS) of KNMI/ESA (http://www.temis.nl, last access: 4 March 2023).

Daily ozone columns at the stations mentioned in Table 1 are retrieved from the global gridded MSR2 assimilated data fields. Data corresponding to the grid cell with forecast error estimate higher than 20 DU for MSR2 were removed following

indications given on the TEMIS/ESA web site. This filter was increased to 35 DU for 1993-1994 in the Southern Hemisphere (SH) to allow data in July for normalisation (see Section 3, Methodology).

### 2.3 TOMCAT/SLIMCAT model

The three-dimensional off-line CTM TOMCAT/SLIMCAT (Chipperfield, 1999) (hereafter SLIMCAT) is used. to simulate passive odd-oxygen tracer that is transported/advected without any interactive chemistry (Feng et al., 2005), and active ozone

with full stratospheric chemistry including heterogeneous reactions on sulphate aerosols and polar stratospheric clouds (PSCs) (Feng et al., 2021). In this study, SLIMCAT is forced by wind and temperature fields from the European Centre for Medium-Range Weather Forecasts (ECMWF) ERA5 reanalysis (Hersbach et al., 2020). The model uses a hybrid σ-pressure as vertical coordinate. The tracer advection uses the conservation of second-order moments scheme by Prather (1986). The vertical transport is diagnosed using mass flux divergence (Chipperfield, 2006).

The long-term simulations used in this work start in 1980 (Feng et al. 2021) with a horizontal resolution of 2.8° latitude × 2.8° longitude and 32 vertical levels from the surface to ~65 km. The passive ozone tracer is reinitialized each year on July 1[st] in the Southern Hemisphere (SH) and December 1[st] in the Northern Hemisphere (NH) by setting it equal to the modelled active chemical ozone field. The passive and active ozone columns are sampled above the stations of Table 1 at 12 UT. The SLIMCAT model has been widely used in previous studies of stratospheric ozone (e.g. Feng et al., 2021).

### 3 Methodology

The ozone loss is obtained by applying the passive tracer method (Goutail et al., 1999) which has been applied in different studies to calculate ozone loss in the SH (e.g. Kuttippurath et al., 2010; 2013) and the NH (e.g. Pommereau et al., 2013; 2018) using MSR2 or SAOZ data. The loss is computed at each station of Table 1 by subtracting the measured total ozone (SAOZ and MSR2 merged dataset, hereafter called OBS) inside the polar vortex from the corresponding passive ozone column

simulated by SLIMCAT. To determine if the station is inside the vortex, the Nash et al. (1996) criterion is applied on the Equivalent Latitude (EL) – isentropic level (θ) quasi-conservative coordinate system (McIntyre and Palmer, 1984). This system can be assimilated to 2-D vortex-following coordinates where the pole corresponds to the position of maximum potential



vorticity (PV). The wind and temperature fields from ERA5 reanalysis are used to calculate the 2-D coordinate system. The vortex edge is considered as the limit between a region inside and outside the vortex, corresponding to the EL of maximum

PV gradient, weighted by the wind module temporally smoothed with a 5-day moving average, as described in Pazmiño et al. (2018). In this work, the classification of the station with respect to the position of the vortex is considered at the 475 K isentropic level (~18 km), where the ozone maximum is observed in winter/spring and as used in previous works (e.g., Kuttippurath et al., 2010; Pommereau et al., 2013).

Before the subtraction, the SLIMCAT passive ozone tracer is normalized to the MSR2 ozone dataset. The normalization

coefficient is calculated at each station considering the difference between the monthly mean values of the MSR2 and SLIMCAT active ozone tracer in December (July) for the NH (SH). SAOZ measurements are also normalized by the mean difference between MSR2 and SAOZ data at the beginning of each winter (December/July for NH/SH), or if not available at high latitudes, then in March (August), in the NH (SH). The amplitude of the mean monthly difference between MSR2 and normalized SAOZ data is less than 1% during the winter which is smaller than SAOZ error bars.

Figure 1 shows the evolution of MSR2 and SAOZ ozone observations and normalized ozone columns (both passive and active) from the model at Ny-Alesund during the Arctic winter 2021/2022. The top panel shows the position of the station and the vortex edge on the equivalent latitude scale at the 475 K isentropic level. The SLIMCAT tracer captures the short-term ozone fluctuations resulting from horizontal and vertical transports linked to the propagation of the planetary waves. The horizontal transitions between regions inside and outside the vortex are observed by mid-March (day ~70) with ozone values increasing

from ~300 DU to ~550 DU. The progression of chemical ozone loss (100*(passive tracer – OBS)/passive tracer) above the station is observed to reach 112 DU on Julian day 83, corresponding to about 23% (Fig. 1, bottom panel). The agreement observed between the MSR2 and SAOZ datasets after normalization gives confidence in this simple method to build the OBS merged dataset). The mean bias between MSR2 and the normalized SAOZ datasets in the NH are within ±0.3 DU at each station and in the SH between 0 and -1 DU with a standard deviation of the mean lower than 1 DU.



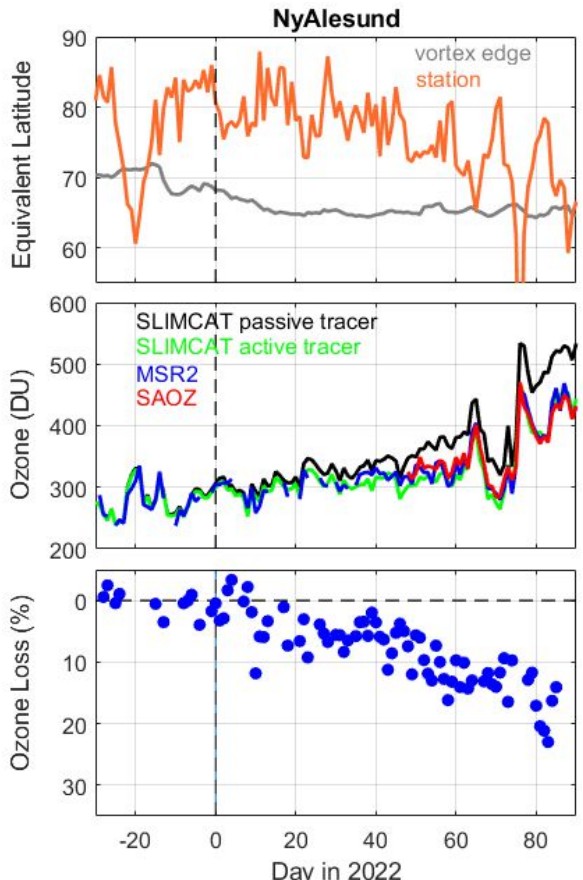

**Figure 1. Top panel: evolution of the position of the vortex edge over Ny-Alesund station in equivalent latitude scale at the 475 K isentropic level. Middle panel: evolution of ozone columns at Ny-Alesund from reanalysis MSR2 fields, SAOZ observations and simulated by SLIMCAT. Bottom panel: evolution of ozone loss (in %) at Ny-Alesund derived from OBS merged dataset (see the text) and the SLIMCAT passive tracer.**

The relative ozone losses at each station (Table 1) within the vortex are considered altogether and a 10-day running median is applied during the winter. Figure 2 shows the evolution of the relative ozone loss during the 2022 NH winter (black line) obtained from the ozone loss values above the different stations (symbols in colour). At the end of the winter, the accumulated ozone loss is considered as complete when temperatures within the vortex are higher than the temperature threshold for nitric acid trihydrate (NAT) PSC formation ($T_{NAT}$). At that time, the diurnal $NO_2$ difference rapidly increases and ClO values from SLIMCAT rapidly decrease (not shown). During the 2022 NH winter, a fast increase of the diurnal $NO_2$ difference is observed after day 60 (as shown in the bottom panel of Figure 2), as a signature of chlorine deactivation. Long periods were also observed with minimum temperature lower than $T_{NAT}$ inside the vortex during 105 and 81 days at the 475 and 550 K isentropic levels, respectively, as shown in Fig. 3 (top panel). The considered thresholds are 195 K and 192 K for the 475 K and 550 K isentropic levels, respectively (Pommereau et al., 2013). PSC formation stops first at the higher levels on day 50 and then later on day 75



at the lower levels. The accumulated ozone loss observed on day 80 reaches 18.1±0.5 % (87±2.7 DU). The standard error of the median corresponding here to the half of the Q84-Q16 or 68% interpercentile spread (IP68) is also shown.

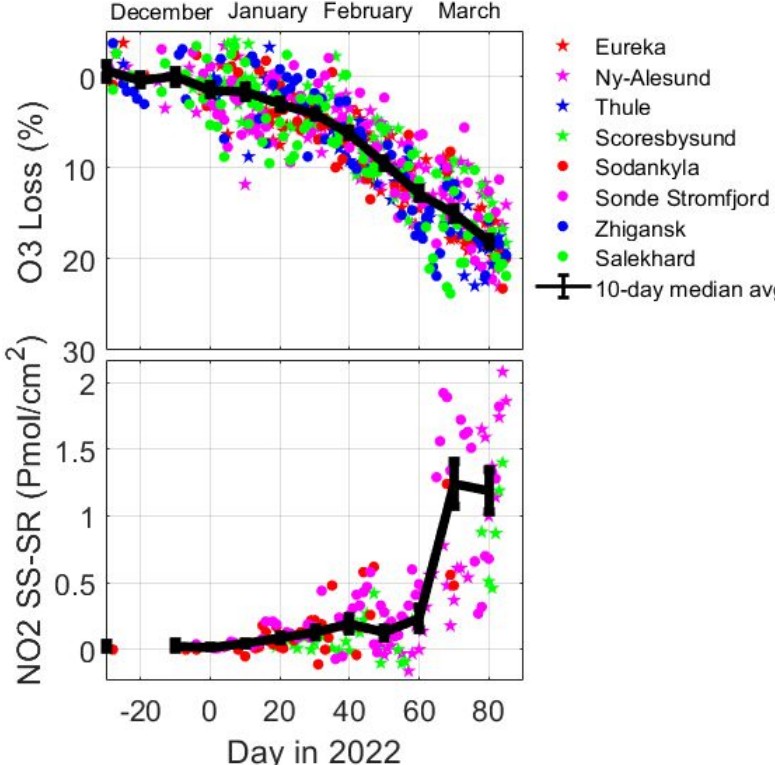

**Figure 2. Top panel: time series of observed ozone loss (%) inside the vortex above each SAOZ station for the 2022 NH winter.**

**Bottom panel: Time series of the amplitude of the NO₂ diurnal variation inside the vortex above SAOZ stations. The 10-day running median and standard error of the median (IP68/2, see the text) are superimposed by the black line on both panels.**



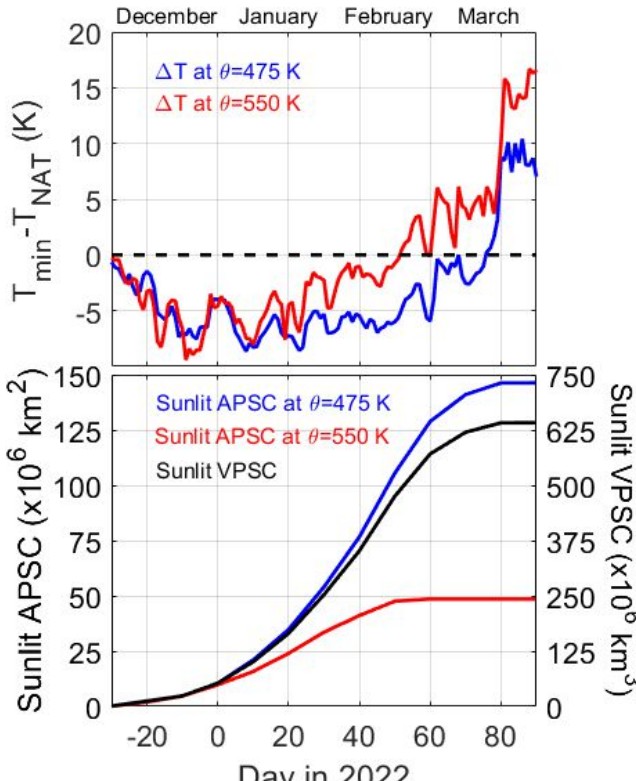

**Figure 3. Top panel: time series of difference between minimum temperatures and T$_{NAT}$ (K) at 475 and 550 K isentropic levels for the 2022 NH winter. Negative values correspond to the period of PSC formation at the corresponding isentropic level. Bottom panel: cumulative time series of sunlit areas of PSC at 475 (blue) and 550 K (red). The sunlit volume of VPSC computed following Rex et al. (2004) is superimposed by a black curve.**

The interannual behaviour of ozone loss related to PSCs, which plays a crucial role on ozone polar heterogeneous chemistry is also analysed. The cumulative surface of the polar vortex exposed to temperatures lower than the NAT PSC formation threshold coincident with sunlit regions (SZA<93°) was computed at 475 K and 550 K. This cumulative surface is hereafter referred to as sunlit APSC$_\theta$. The APSC on the 475 and 550 K isentropic levels are shown in Fig. 3 (bottom panel) for the 2022 NH winter. The sunlit NAT PSC volume (sunlit VPSC) was estimated following the relationship of Rex et al. (2004) and integrated through the end of the winter. The sunlit VPSC is considered as a proxy of chlorine activation. The computed VPSC for the 2022 NH winter is superimposed on the bottom panel of Fig. 3 (black curve).

## 4 Polar ozone loss in the 2018-2022 period

Since 2000, an increasing interannual ozone loss variability is observed in both hemispheres, particularly in the SH, compared to previous winters. Figure 4 presents the evolution of ozone loss calculated by our method between 2018-2021 in the SH (left



panel) and 2018-2022 in the NH (right panel). Ozone losses in previous atypical years are also shown (dotted lines), e.g. the NH 2011 record ozone loss (Pommereau et al., 2013) that was due to a cold, strong and long-lasting polar vortex (Manney et al., 2011) and the 2002 SH weak ozone loss (Hoppel et al., 2003) linked to unprecedented large wave activity (Allen et al., 2003) resulting in a major sudden stratospheric warming (SSW) and a split of the vortex in the middle stratosphere at the end of September. The median values of ozone loss for the 1989-2017 winters in the SH, the 1990-2017 winters in the NH and the corresponding 20th and 80th percentiles are also represented in Figure 4 by the black lines and shaded area, respectively. Similar to Fig. 4, the $T_{min}$-$T_{NAT}$ anomaly at 475 K is shown for the last winters in Fig. 5, the 45-day mean heat flux in the 45°–75° latitude range at 70 hPa from MERRA-2 analyses (NASA's Goddard Space Flight Center https://acd-ext.gsfc.nasa.gov/Data_services/met/ann_data.html, last access: 6 October 2022) in Fig. 6 to evaluate the impact of dynamical activity. Figure 7 plots the proxy GRAD corresponding to the maximum gradient of PV as a function of equivalent latitude within the vortex boundary region (Pazmiño et al., 2018) to evaluate the stability of the vortex during the study period. Pazmiño et al. (2018) used both proxies to characterize the interannual evolution of total ozone in Antarctica during the September and October periods

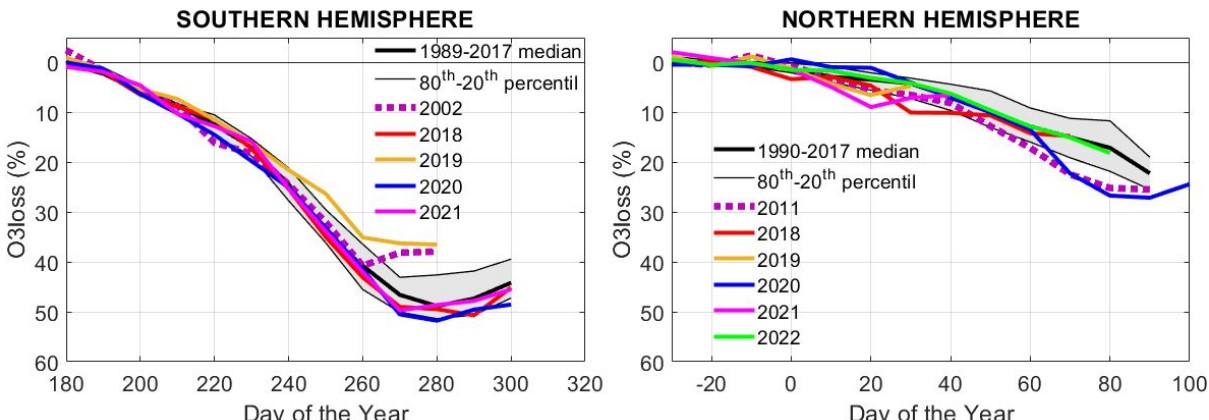

**Figure 4. Evolution of ozone loss in recent winters using the merged OBS dataset: SH 2018-2021 (left panel) and NH 2018/19 – 2021/22 (right panel). Unusual winters are also represented by pink dashed lines: weak ozone loss in SH (2002) and 2011 record ozone loss in NH. The median and 20th—80th percentile climatological values of previous winters are represented by thick and thin black lines, respectively.**

The median value of the accumulated ozone loss at the end of the winter is more than 2 times larger in the SH than in the NH. The recent winters present an accumulated ozone loss varying from 7 to 27% in the NH and 37 to 52% in the SH. The interannual variability of the ozone loss represented by the maximum amplitude of ozone loss between the recent winters is mostly similar in both hemispheres: 20% in the NH and 15% in the SH.



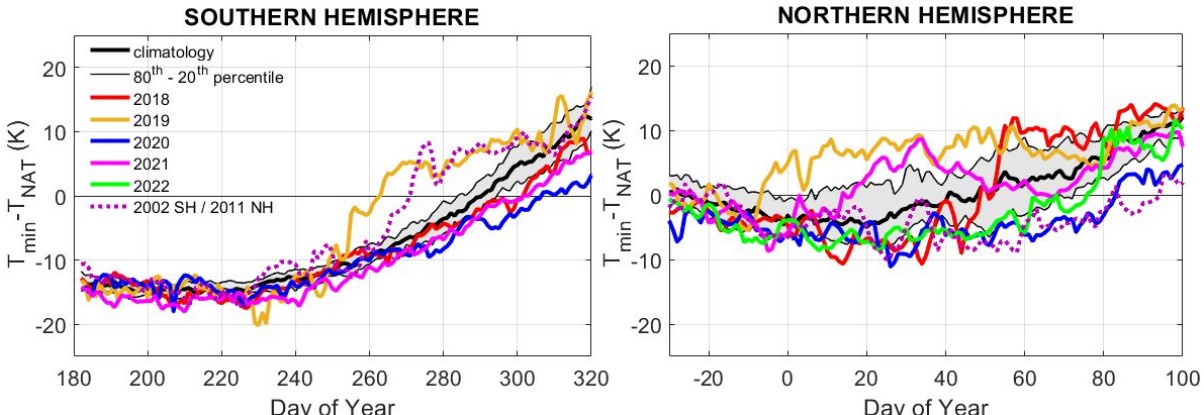

**Figure 5. As Fig. 4 but for temperature anomaly at 475 K using ERA5 reanalyses.**

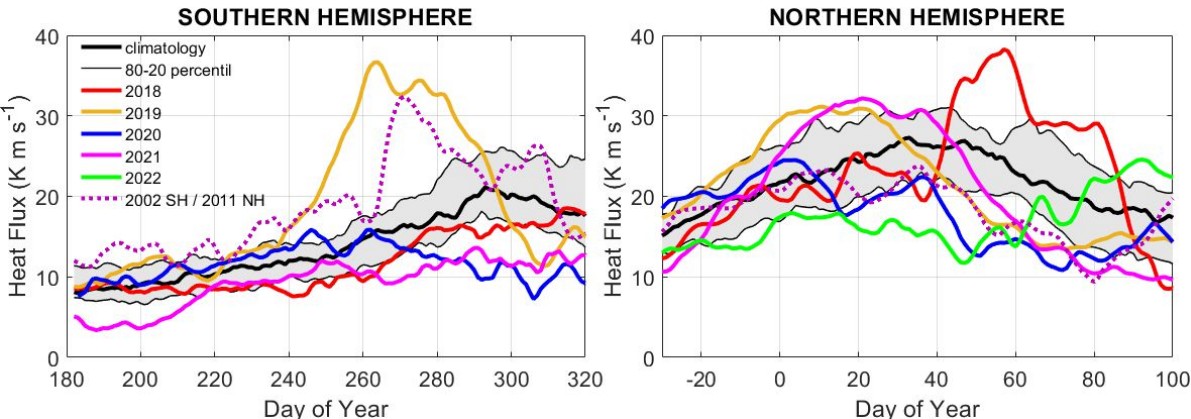

**Figure 6. As Fig. 4 but for 45-day mean heat flux in the 45°–75° latitude range at 70 hPa from MERRA-2 analyses.**

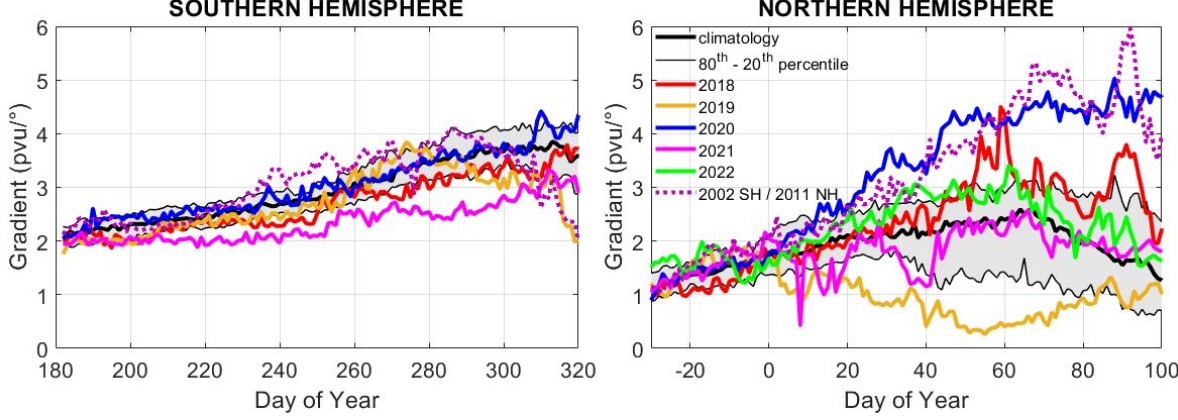

**Figure 7. As Fig. 4 but for the PV gradient in the vortex edge as defined in Pazmiño et al. (2018) using ERA5 reanalyses.**



### 4.1 Southern Hemisphere

In the SH (left panel of Fig. 4), the evolution of the ozone loss during the recent winters is found to be within the climatological values (grey area) until the end of August (day 240). For the four years shown, temperatures lower than $T_{NAT}$ are observed early, from mid-May at the 475 K isentropic level.

The 2018 SH winter is a typical one (cold winter with a strong vortex) close to the median climatological value reaching a maximum ozone loss of 50.7±1.1% at day 290 (red line). Temperature values lower than $T_{NAT}$ persist until the end of October as shown by $T_{min}$-$T_{NAT}$ anomalies at 475 K in Fig. 5 (left panel). The anomaly is within the 20$^{th}$-80$^{th}$ percentile of the 1989-2017 climatological median values represented by the grey area. The mean anomaly was -10.1±4.8 K for 174 consecutive days. The dynamical activity was near the climatology (Fig 6 left panel) and as well as the vortex stability (Fig. 7, left panel). During the 2019 SH winter, a minor SSW appeared at the end of August linked to wavenumber 1 event, producing a displacement of the vortex at the upper levels and a consecutive decrease of its size. These two factors were similar to the 2002 SH winter (pink dashed line in left panel of Fig. 4) where a major SSW occurred at the end of September, inducing large total ozone values within the vortex (Wargan et al., 2020). In 2019, after the minor SSW at the end of August, the ozone loss started to slow down as shown by a levelling of the ozone loss diverging from the climatological grey area and reaching a maximum of 36.3±1.3% in the first week of October (brown line in Fig. 4). In the case of 2002, the ozone loss stopped rapidly after the major warming and reached a slightly larger ozone loss compared to 2019. The period with temperatures lower than $T_{NAT}$ was reduced by 1 month compared to 2018 and displayed a mean T anomaly slightly higher than in 2018 (-11.4±4.2 K during 132 consecutive days). The dynamical activity is well represented in Fig. 6, where the heat flux rapidly increases at the end of August with values much higher than climatology until mid-October and comparable to the NH (Fig. 6, right panel). The stability of the vortex was within the climatological values until mid-October, slowing down rapidly after.

The SH stratosphere in 2020 was strongly impacted by the enhancement of aerosol levels from the severe south-eastern Australia bushfires during 29 December 2019 to 4 January 2020, known as the Australian New Year (ANY) fires (e.g. Khaykin et al., 2020). Rieger et al. (2021) showed ozone negatives anomalies in mid-latitude and polar regions from OMPS satellite observations linked to the ANY event of similar magnitude related to the Calbuco volcanic eruption in April 2015 in the south of Chile. In the Antarctic, the 2020 winter ozone loss evolution is within climatological values until the end of September (blue line in Fig. 4). Temperatures lower than $T_{NAT}$ are already present in May until the beginning of November (176 days) with a mean T anomaly of -10.1±4.5 K as in 2018 but a much larger sunlit VPSC. The maximum ozone loss of 51.8±1.4% was found in early October, a value outside the 20-80% percentile range of the climatology. The persistently cold temperature in the lower stratosphere in the polar region in 2020, led in October to an acceleration of the ozone loss and a delayed break-up of the polar vortex, explaining the long-lasting ozone loss during the months of October to November (Damany-Pearce et al., 2022). The heat flux exhibits values within the climatology until the end of September before slowing down rapidly during October (Fig. 6) and the strength of the vortex edge was close to the median climatological value (Fig. 7).



During the 2021 SH winter, the evolution of ozone loss was within the climatological values until the end of August (day 240). The temperatures lower than $T_{NAT}$ started later than previous years in May, persisting until the end of October as in 2018 and 2020 with a T anomaly of -11±5.3 K corresponding to 167 days (Fig. 5). The ozone loss ratio increases between the end of

August and beginning of October (day 270), reaching the lower limit of climatological ozone loss values. The sunlit VPSC are similar to those of 2018, but the strength of the vortex is weaker than in previous years as shown by the low PV gradient in Fig. 7. This year presents also lower heat flux values than the climatology before August and after mid-September (Fig. 6) and the final accumulated ozone loss reaches 49.7±0.9% (pink line in Fig. 4) within the climatology.

**4.2 Northern Hemisphere**

In the NH (right panel of Fig. 4), the evolution of the accumulated ozone loss is strongly dependent on the temperature history. The ozone loss already starts to vary from one year to the next in December. Perturbed winters due to enhanced wave activity could favour mixing across the polar vortex.

The 2018 NH winter (red line in the right panel of Fig. 4) displays higher ozone loss than the 20th-80th interpercentiles (grey area) from mid-December to mid-February. Temperatures within the vortex at the 475 K isentropic level were much lower

than the $T_{NAT}$ threshold from early December until mid-February, with a mean $T_{min}$ anomaly of -5.3 K (Fig. 5, right panel). The major SSW on 12 February linked mainly to wavenumber-2 forcing (Butler et al., 2020) induced a rapid increase of temperature and a split of the vortex. This increase in dynamical activity is also highlighted by the increase of the heat flux (Fig. 6). The strengthening of the vortex presented values higher than climatology (Fig. 7). The very low temperatures for the remaining ~80 days within the vortex allowed moderate ozone loss of 14.7±0.8%.

The 2019 NH winter also presented a SSW but early in the year corresponding to the January single warming mode (Mariaccia et al., 2022), as shown by the increase of heat flux outside the climatological values at the end of December (brown line in Fig. 6). The major SSW of 2 January 2019 was linked to a wavenumber-1 event (Butler et al., 2020). The vortex weakened more rapidly after the SSW and remained at low values thereafter (Fig. 7). Temperatures lower than $T_{NAT}$ were observed during 20 (non-consecutive) days in December with a mean anomaly value of -2.2 K at the 475 K isentropic level (Fig. 5). The

accumulated ozone loss of 2019 warm winter was 6.5±1.4% (Fig. 4).

The 2020 NH winter is associated with record-low ozone values within the vortex which are explained by a long period of temperatures lower than $T_{NAT}$ from December to mid-March (113 days at 475 K isentropic level, blue line in Fig. 5), a large stability of the vortex (Fig. 7) and a low ozone resupply from lower latitudes (e.g. Manney et al., 2020). In the beginning of December, temperature anomalies were near -4 K and the mean value of the whole winter reached -5.3 K as in 2018. The

ozone loss was within the climatological values until March, but a rapid increase of 13% during March led to an accumulated ozone loss of 27.1±1.1% (Fig. 4). Mariaccia et al. (2022) classified this winter as an unperturbed radiative final warming mode as shown by the low values of the heat flux in Fig. 6. Comparing the 2020 and 2011 winters with pronounced ozone loss (pink dashed line in Fig. 4), we find a similar maximum ozone loss at the end of March, which is due to the persistent cold temperatures lower than $T_{NAT}$ for ~110 days (Fig. 5), a weak dynamical activity (Fig. 6) and a strong vortex (Fig. 7).





The 2021 NH winter experienced a major SSW on 5 January (pink line in Fig. 5). Temperatures lower than $T_{NAT}$ were observed during 41 consecutive days between early December and mid-January with a mean value of T anomaly of -3.4 K (Fig. 6). During this period a rapid ozone loss evolution outside the climatological values is observed at the beginning of January slowed down by the SSW event that stopped it on January 20 (Fig. 4). The accumulated ozone loss was only 8.9±1.2%.

The 2022 NH winter is associated with an unperturbed dynamical final warming mode as shown by the low values of heat flux
until beginning of March (day 60) (green line in Fig. 6). It was a cold and long-lasting winter with temperatures lower than $T_{NAT}$ until mid-March (105 days at 475 K, Fig. 5) with a mean value of -6.5 K for the T anomaly. The ozone loss is well within the climatological values with an accumulated ozone loss of 18.1±0.5% (Fig. 4).

## 5 Long-term evolution of ozone loss

In order to study a possible recovery of total ozone columns in the polar regions, three different metrics were applied to the
ozone loss datasets. Then a robust linear fit was calculated since 2000, the year of maximum ODS amounts in the polar stratosphere (WMO 2014).

### 5.1 Maximum ozone loss

The first metric considered is the maximum ozone loss (MOLoss) for each winter, which corresponds to the maximum value of the accumulated ozone loss within the respective winter period as considered in Section 3. Figure 8 shows the interannual
evolution of MOLoss for both hemispheres (coloured lines). The model results using its active tracer are also represented (grey lines). A good agreement is observed in the interannual variability of observations and simulations in both hemispheres, with systematically smaller values in the simulations since 2003 in the SH. As expected, the NH MOLoss shows smaller values but larger interannual variability, which is intrinsically linked to a more disturbed stratospheric dynamic.

In the SH, a stabilization of the MOLoss is observed in the 1990s at about 50% and a slight decrease since 2000 with an
enhanced interannual variability in the last decade. A similar negative trend in ozone loss is found based on observed (OBS) and modelled results, with values of -2.3±1.5% and -2.9±1.5% dec[-1] respectively. The trends are significant only at 1 standard error ($\sigma$). In particular, the SSW years 2002 and 2019 are characterised by smaller MOLoss values, followed by 2004, 2012, 2013 and 2017. The years 2002, 2004, 2012 and 2013 were identified by Lim et al. (2019) as years of weak SH polar vortex. In 2017, the heat flux (not shown) presents values higher than the climatological envelope from the end of August to the end
of September, with T anomalies rapidly increasing by 8 K with respect to the median in the second half of September, which could have slowed down the chemical ozone depletion.



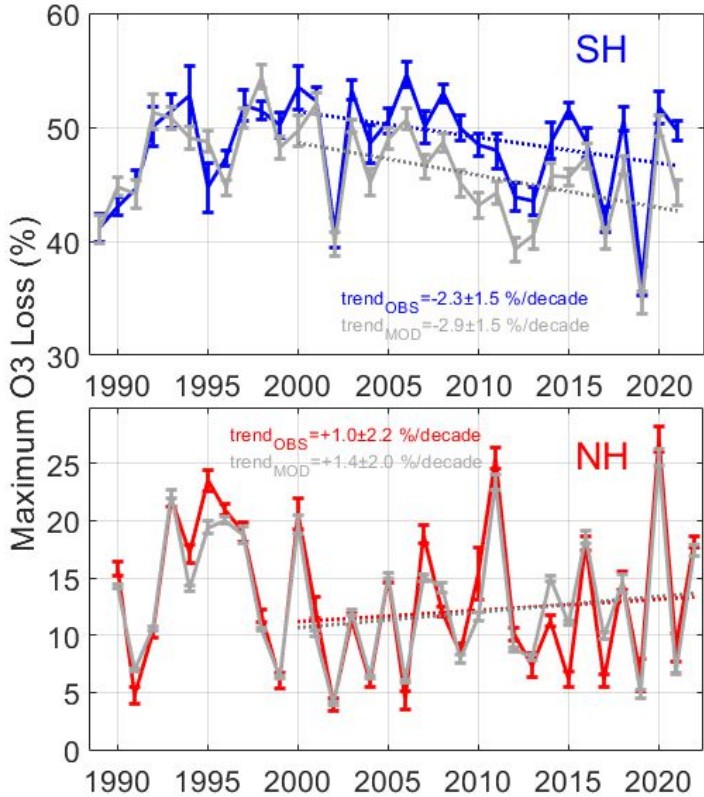

**Figure 8. Interannual evolution of the maximum ozone loss obtained using the passive tracer method and the merged OBS dataset (colour lines) and model (gray line) in the SH (top panel) and NH (bottom panel). The estimated robust trends since 2000 were added to the figures with the corresponding colour codes.**

In the NH, the average MOLoss is less than half of that observed in the SH. The large ozone losses in the mid-1990s NH are shown in Fig 8 bottom panel with values near 20%. There is substantial interannual variability between warm and cold winters with two record values of ozone loss in 2011 and 2020. The trend values estimated since 2000 are positive towards $1.0\pm2.2\%$ dec$^{-1}$ but they are not significant. This metric does not allow the detection of any trend in the NH.

**5.2 Ozone loss onset day**

The ozone loss onset day (OLossOnset) metric was developed to analyse the evolution of the ozone loss at different thresholds values, expecting later onset of polar ozone loss in relation with lower amounts of ODS in the stratosphere. The onset day is determined as the day when the 10-day running median ozone loss crosses a determined threshold value. A similar metric for total ozone values inside the vortex was used in a previous study (Pazmiño et al., 2018). In this study, the ozone loss time dataset is used instead of total ozone columns in order to consider chemical processes only. Figure 9 presents the evolution of OLossOnset at five different thresholds of ozone loss for SH (left panel) and NH (right panel).



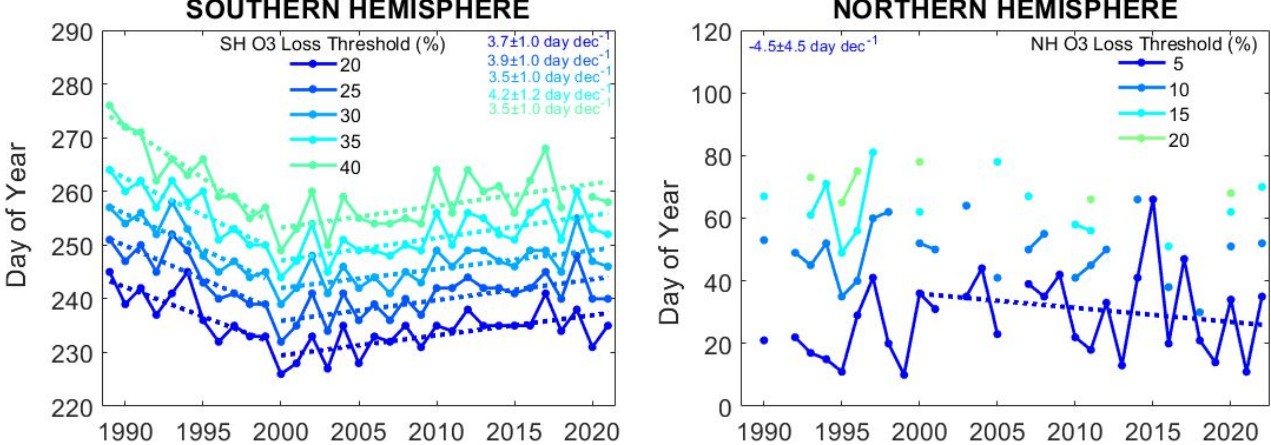

**Figure 9. Onset day when 10-day averaged ozone loss reach a particular ozone loss value: 20, 25, 30, 35 and 40% for the SH (left panel) and 5, 10, 15 and 20% for NH (right panel). Robust linear fits before and after 2000 are also shown for the SH (dashed lines).**

In the case of the SH, the chosen ozone loss threshold values enable a long-term estimation of the interannual evolution of OLossOnset. The trend estimations were performed before and after 2000. All trends estimated by independently robust linear regression are significant at least at $2\sigma$. The lower trend values are observed for the threshold of 20% and the highest ones for 40% of ozone loss before and after 2000. The positive trends vary between $3.5\pm1.0$ and $4.2\pm1.2$ day dec⁻¹. The ratio between the trends before and after 2000 of each OLossOnset dataset is -0.3, with the exception of the threshold of 40% where -0.2 is found due to the steeper slope observed before 2000.

For the NH, only the OLossOnset at the threshold of 5% is reached almost each year of the considered period. The trend observed is marginally significant (-4.5±4.5 day dec⁻¹). The other thresholds do not allow any robust statistical analysis. This metric does not allow the detection of any trend in the NH.

### 5.3 Residuals of ozone loss/VPSC relationship

Figure 10 presents the ozone loss value as a function of sunlit VPSC for each winter of the NH (triangles) and SH (inverse triangles). The figure highlights the difference between both hemispheres with much higher sunlit VPSC in the SH and consequently higher ozone loss. The dynamical range of sunlit VPSC in the SH varies between $2 \times 10^9$ and $5 \times 10^9$ km³, which corresponds to an ozone loss between 36 and 55%. The dynamical range of sunlit VPSC in the NH is much smaller ($< 10^9$ km³) but the dynamical ozone loss range is slightly higher (4-27%). A 3rd-order polynomial was applied to represent the relationship between ozone loss and sunlit VPSC. In the figure, most years of the last decade present smaller ozone loss values with respect to the polynomial fit (yellow to red colours). The figure highlights a quasi-linear relationship between ozone loss and VPSC in the NH (lower-left region in Fig. 10) and a different behaviour for larger ozone loss values due to saturation of ozone loss in the lower stratosphere in the SH (e.g., Yang et al., 2008). Therefore, a fit can be applied independently in each




hemisphere (linear for the NH and parabolic for the SH due to the saturation of ozone loss in the lower stratosphere). The thick black lines in Fig. 10 represent the corresponding fit (fit_O3Loss(sunlitVPSC)) for each hemisphere.

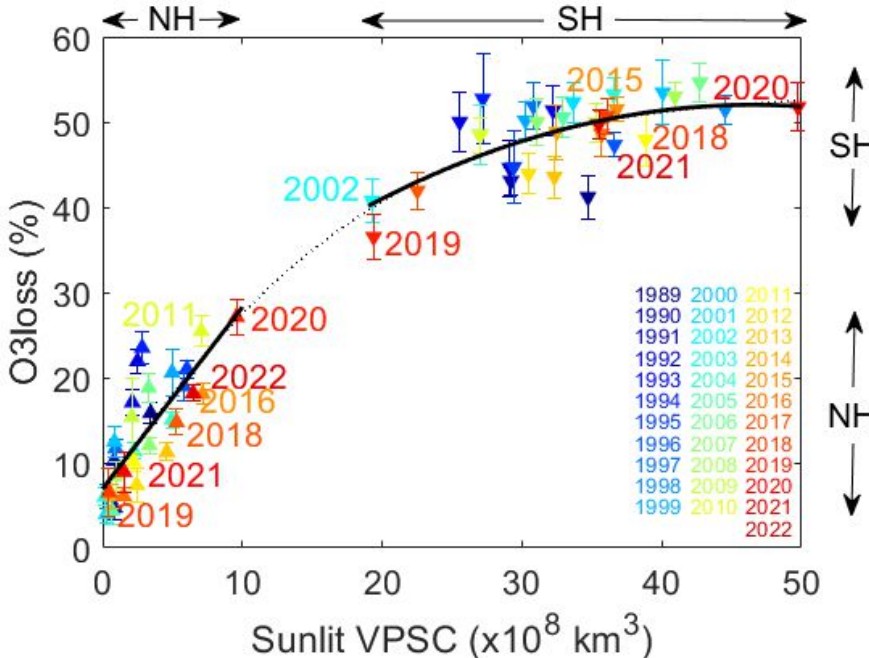

**Figure 10. Ozone loss as a function of sunlit volume of polar stratospheric clouds (VPSC) for each winter for the Northern and**
**Southern hemispheres. The 68% interpercentile of ozone losses are also represented (see Sect. 3, Methodology). The colour code represents the years. The linear and parabolic fits are represented for the NH and SH, respectively, (black lines) and the 3rd-order fit considering years for both hemispheres is represented by the black dotted line.**

The residual of the ozone loss with respect to the linear (NH) or parabolic (SH) fit (ROLoss) is calculated for each year of the
corresponding hemisphere as follows:

ROLoss(year)=O3Loss(year)- fit_O3Loss (sunlitVPSC (year))                             (1)

Figure 11 shows the ROLoss dataset for the SH (top panel) and NH (bottom panel) respectively. In both panels, residuals vary approximately between -5 and +10%. Larger positive residuals are observed in the 1992-1995 period. Afterwards a decrease is observed starting in 1996 in both hemispheres with a higher interannual variability in the NH. A robust linear fit was applied
to both datasets since 2000 to evaluate a possible recovery.



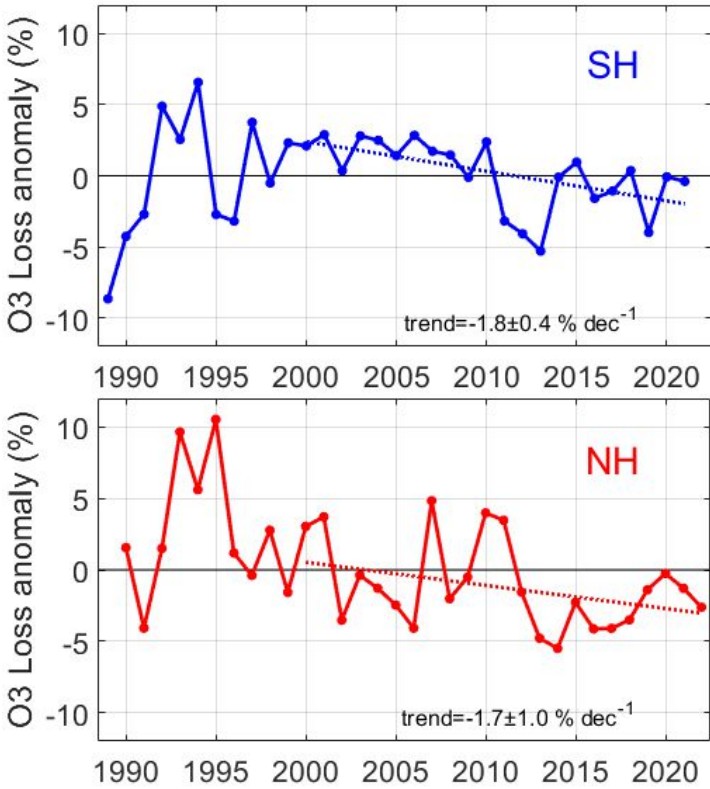

**Figure 11. Interannual evolution of Residuals of Ozone Loss with respect to regressed ozone loss values computed following Eq. 1 for the SH (top panel) and NH (bottom panel). Estimated trends since 2000 are represented by dotted lines.**


In the SH, a negative trend of the ozone loss residuals of -1.8±0.4% dec$^{-1}$ is observed, significant at $2\sigma$. Most years match well with the trend line with the exception of some years characterized by a weaker vortex or high dynamical activity such as 2002, 2012, 2013, 2019 and 2021.

In the case of the NH, a negative trend is also observed but only significant at $1\sigma$ (-1.7±1.0% dec$^{-1}$). The interannual variability is large but the standard error of the slope is similar in both hemispheres. Since 2012, the residuals are mostly negative and only the record year of 2020 presents a slightly positive (SH) or null (NH) value. Unlike the other two metrics, this metric does allow the detection of a trend in the NH.

## 6 Conclusions

Ozone loss datasets extending for more than 30 years were computed for both polar regions using the passive ozone tracer values simulated by the SLIMCAT CTM combined with SAOZ ground-based data merged with the MSR-2 reanalysis. Although the passive tracer method enables the identification of ozone evolution only due to chemistry, this chemistry can be





influenced by dynamical processes via their effect on temperature. The ozone loss shows a linear relationship with sunlit VPSC within the vortex for the NH and a parabolic behaviour in the SH due to the saturation effect of the ozone loss in the Antarctic stratosphere.

The analysis of ozone loss in the polar winters since 2018 shows that much of the loss lies between the 20th and 80th percentiles of the values observed in previous years and that they are well correlated with the temperature history (Fig. 4 and 5). The extreme years are prominent in the ozone loss datasets with 1) an atypical weak ozone loss in the 2019 SH caused by an early minor SSW at the end of August due to the strong dynamical activity in that year, comparable to what is generally observed in the NH (Fig. 6); and 2) a large ozone loss in 2020 in both hemispheres with 7% higher values than the median climatology

and linked to very cold and long-lasting winters. Notably the strength of the vortex edge in the 2020 NH is larger than the values observed in the SH climatology including year 2020 (Fig. 7).

In order to estimate a possible recovery of ozone, trends since 2000 were computed for three different metrics. In the first case, based on the maximum ozone loss found at the end of the winter, a negative trend of -2.3±1.5% dec$^{-1}$ was found in the SH, only significant at $1\sigma$. In the NH, a positive trend of +1.0±2.2% dec$^{-1}$ was calculated but was not significant. This positive

trend is mostly influenced by the record ozone loss years 2011 and 2020. Regarding the SH, this metric appears sensitive to dynamics since the maximum in ozone loss generally occurs between days 270-290, in October, a month characterized by higher temperatures within the vortex and larger transport variability (Solomon et al., 2016). The second metric takes into account the interannual evolution of the onset day when the ozone loss reaches different thresholds, similar to the methodology developed for total ozone values by Pazmiño et al. (2018). In the SH, this metric shows a positive trend of +3.8±1.0 day dec$^{-1}$

on average. This trend is significant at $2\sigma$ and could be related to the lower ODS amounts in the polar austral stratosphere compared to the period before 2000. The various thresholds are reached in September, so this metric is sensitive to the ozone loss at a time that is less affected by dynamical processes compared to October when the maximum ozone loss is reached. In the NH, this metric does not show a statistically significant trend, due to the large interannual variability and the fact that most of the thresholds are not reached in the period studied. The third metric takes into account the relationship between ozone loss

and the sunlit volume of PSCs, linked to heterogeneous chemical processes. In the SH, the ozone loss residuals show a negative trend since 2000 of -1.8±0.4% dec$^{-1}$ significant at $2\sigma$, indicating a statistical significant signal for the recovery of ozone. This value is close to that obtained with the first metric. In the case of the NH, for the first time, a recovery is observed based on this metric, which displays a trend of -1.7%.dec$^{-1}$, significant only at $1\sigma$. Note that this trend value has a similar in value to that in the SH.

In conclusion, our study confirms the ozone recovery in the SH, significant for two of the three metrics based on the ozone loss datasets despite the higher interannual variability in the last decade. In the NH, our study shows for the first time a decrease of ozone loss with respect to sunlit VPSC within the Arctic vortex but only significant at $1\sigma$. In the same way, Bernet et al. (2023) applied the linear regression model from the Long-term Ozone Trends and Uncertainties in the Stratosphere (LOTUS) project only on datasets from three high-latitude stations (Oslo, Andoya and Ny-Alesund) and found positive trends of around



3 % dec$^{-1}$ in March for the 2000-2020 period. However these trends are also not significant at $2\sigma$. Considering the interannual variability in the NH and the associated uncertainties in the ozone loss versus sunlit VPSC regressed values, more years of observations are needed to confirm the trend and to quantitatively attribute the decreasing total ozone loss trend to reductions in ozone-depleting substances.

**Data Availability**

SAOZ data can be obtained through the NDACC database (https://www-air.larc.nasa.gov/missions/ndacc/) and the SAOZ webpage (http://saoz.obs.uvsq.fr/).

The MSR2 data is publicly available at TEMIS webpage of KNMI/ESA (http://www.temis.nl).

ERA5 reanalyses were provided by ESPRI data centre of Institut Pierre Simone Laplace (IPSL).

Model simulations of TOMCAT/SLIMCAT and OBS merged dataset used in this article are available in the following depository: https://doi.org/10.5281/zenodo.7847522 (Pazmino, et al., 2023).

45-day mean heat flux dataset of MERRA-2 is available from NASA's Goddard Space Flight Center webpage (https://acd-ext.gsfc.nasa.gov/Data_services/met/ann_data.html)

**Author contribution**

AP, FG, JPP & FL conceived the study. AP & FG performed the ozone loss analysis. AP constructed the different metrics and computed trends with scientific insight of SGB & AH. MPC & WF performed the model runs. AL contributed to elaborate the GRAD proxy. MVR, NS, GH, RK, KS, KAW & SC provided SAOZ data. The paper was written by AP with contributions of all co-authors.


**Competing interests:** The authors declare that they have no conflicts of interest.

**Acknowledgements**

The authors warmly thank the Institut National des Sciences de l'Univers (INSU) of the Centre National de la Recherche

Scientifique (CNRS), the IPEV and the Centre National d'Études Spatiales (CNES) for supporting the observations of the SAOZ instruments of the French ACTRIS Infrastructure. The authors thank technical teams operating SAOZ instruments. The SAOZ measurements at Eureka were made at the Polar Environment Atmospheric Research Laboratory (PEARL) by the Canadian Network for the Detection of Atmospheric Change (CANDAC), primarily supported by the Natural Sciences and Engineering Research Council of Canada, Environment and Climate Change Canada, and the Canadian Space Agency. The

authors thank TEMIS for total ozone column data of MSR2. They are grateful to Cathy Boone of AERIS/ESPRI/IPSL for providing ERA5/ECMWF data. The TOMCAT/SLIMCAT work at Leeds was supported by NERC projects NE/R001782/1 and NE/V011863/1.



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
