# Peer review of "Trends in polar ozone loss since 1989: Potential sign of recovery in Arctic ozone column"

_EGUsphere, 2023_

## Author Comment (AC1)

**Reply to Anonymous Referee #1 review of manuscript acp-2023-788**

**Trends in polar ozone loss since 1989: First signs of recovery in Arctic ozone column**

**Andrea Pazmino on behalf of all co-authors**

We thank Anonymous Referee #1 for the time devoted to evaluate our work. Your valuable comments have helped us to improve our manuscript. Note that the title of the article and the wording used to explain the results related to metric 3 were changed also in response to referee 2 comments. The new title is:

Trends in polar ozone loss since 1989: **Potential sign** of recovery in Arctic ozone column

Please find our answers below (in red)

**Clarifying questions and comments.**

Line 128. What is the definition of the "overpass" criteria?

Thank you for this question. The overpass values correspond to the averaged amount of the available MSR2 grid values at ±1° of the station coordinates.

The following sentence was modified to clarify the overpass criteria:

"Daily ozone columns at the stations mentioned in Table 1 are retrieved from the global gridded MSR2 assimilated data fields **by averaging the total ozone columns of MSR2 within ±1° of the station coordinat**e".

Line 143. Please provide station and satellite/model matching criteria. The satellite grid is at 0.5 and SLIMCAT model is at 2.8 degrees. Are there additional matching/averaging is done to reduce sampling biases? Also, it might be useful to provide the number of observations for all stations inside of the vortex during the analyzed period.

The SLIMCAT model fields are interpolated linearly in longitude and latitude at each output time during the simulation to obtain profiles at the SAOZ stations. The following sentence in L143 was changed as follows:

"The passive and active ozone columns are sampled above the stations of Table 1 at 12 UT **by performing a bilinear interpolation of the model fields (in longitude and latitude) to the location of the SAOZ stations during the model simulation**."

In order to consider your suggestion about the number of observations inside the vortex, the following text and figure (new Figure 1) was added at the end of the 1ˢᵗ paragraph of the Methodology section:

"**Figure 1 shows the number of merged data inside the vortex for each winter of the considered periods for the SH (blue line) and NH (red line). Between 200 and 400 observations are considered for the Arctic vortex, and about 800 for the Antarctic vortex. The number of the observations in the Arctic vortex displays a large interannual variability while it is much more stable in the Antarctic. These differences are explained by the larger area and the longer persistence of the SH vortex compared to the NH one.**"

[Figure]

**Figure 1. Number of merged data (OBS) inside the vortex for each winter of the SH (blue line) and NH (red line).**

Line 149. Please provide additional details of datasets merger, i.e. temporary and special matching, treatment of missing data, weighting of the MSR2 and SAOZ data in the combined record.

A weighting criterionwas not applied to the MSR2 and SAOZ normalized data since the differences between both data series are lower than respective error bars. The only temporary criteria considered in this work was to select data on the same day in UT and average them. The sentence in L163 was modified as follows:

"**In the case of the days when only one measurement is available, the corresponding value is considered.** The amplitude of the mean monthly difference **during the winter** between **normalized SAOZ data and MSR2 or merged data is less than 2% or 1%, respectively,** which is smaller than the SAOZ **precision (Hendrick et al., 2011**)."

Line 163. What is the reason for not selecting March to normalize all years of SAOZ data? This could make normalization consistent through the entire analysed record.

In the northern hemisphere, normalization of SAOZ data for the four stations equatorward of the polar circle was treated in the same way as the normalization of passive and active tracers of the SLIMCAT model in the different stations. Since the ozone loss is a value relative to that in the beginning of the winter, a good normalization during that period is essential. The additional data available since February-March were considered to improve the sampling within the vortex and a simple normalization in March was used to avoid any large bias. The mean differences in March between merged data (OBS) and SAOZ at the four stations vary between 0.15% to 1.1%, which is lower than the error bars of ozone loss at the end of the winter.

Line 185. Please clarify what you mean by "diurnal differences".

In the sub-section 2.1 the diurnal difference was specified in L115

"The difference between sunset and sunrise $NO_2$ total columns is calculated at each SAOZ station to follow the amplitude of the $NO_2$ diurnal cycle"

We have changed the sentence in L185 as follows

"At that time, the diurnal $NO_2$ difference rapidly increases **(Fig. 3, bottom panel)** and ClO values from SLIMCAT rapidly decrease (not shown)."

And the legend of the Fig. 2 (new Fig. 3)

"Figure **3**. Top panel: time series of observed ozone loss (%) inside the vortex above each SAOZ station for the 2022 NH winter. Bottom panel: Time series of the amplitude of the $NO_2$ diurnal variation (**NO₂ sunset – NO₂**

**sunrise)** inside the vortex above SAOZ stations. The 10-day running median and standard error of the median (IP68/2, see the text) are superimposed by the black line on both panels."

Line 215-216. Can you please mention ozone variability in 2019 that was also an anomalous year in the Antarctic ozone depletion? It clearly deviates from other years.

In this part we only mention the atypical years before 2018. The year 2019 is described in particular in the sub-section 4.1 dedicated to recent years.

Line 228. Fig. 4 caption. I would not say that 2002/2011 winters are unusual anymore since we had similar anomalies in recent years. Do you agree?

We cannot agree from a statistical point of view if we consider only 10% of years as atypical within the last two decades. As can be seen in Fig. 8, there are only two "atypical" years in both hemispheres. 2002 and 2019 stand out as the years with lowest ozone loss in the SH while 2011 and 2020 stand out as the years with largest ozone loss in the NH. Only 2 stratospheric warmings occurred in the SH since 1990: in 2002 and 2019. But indeed, the dynamics play a more important role during the last decade favouring extremes winters.

Lines 325-328. Is there a known reason for the offset between observations and the model since 2003?

For the moment, we do not have any explanation to this difference between model and merged data since 2003. It would be interesting to compare with a long-term run of another CTM model. This analysis could be the objective of a specific work on comparing model simulations but is beyond the scope of this work.

Lines 375-377. Please provide uncertainty of the linear and the parabolic fit for the sunlit PSC area and ozone. What does the SLIMCAT data fit show? Do data and a model fit agree? Can you add a plot that shows the change in the sunlit VPSC as function of time? This could provide a reference of climate change over polar regions.

In order to perform a more robust consideration of the relationship between ozone loss and sunlit VPSC, and to then derive a trend, a multi-parameter regression model was applied to the ozone loss dataset considering as proxies the sunlit VPSC (2nd degree polynomial relationship for the SH and linear relationship for the NH) and a linear trend as a function of time.

The multi-parameter regression was also applied to the ozone loss obtained from the SLIMCAT model simulation. A paragraph at the end of sub-section 5.3 compares the trend using simulations from SLIMCAT to the ones using the merged datasets.

For the uncertainties, please see the answer to the next question which, considering also the comment of Reviewer 2 lead to an update of sub-section 5.3.

Lines 396-402. If uncertainty of the ozone/PSC fit is taken into account, would the trend of the residuals be significant?

A multi-parameter fit of the ozone loss and sunlit VPSC data has been performed since 2000 in order to improve the issues of uncertainties in the regressions. In the Arctic, a trend of $-2.00 \pm 0.97$ % dec$^{-1}$ was found, slightly significant at $2\sigma$. This points to a potential recovery of total ozone in the Arctic. The used multi-parameter fit is explained in detail hereafter.

Thanks to the referees' comments, the sub-section 5.3 was rewritten including now the results obtained by the updated multi-parameter regression model. You will find the new Section 5.3 here below:

"**5.3 Residuals of ozone loss/VPSC relationship**

Climate change can influence the polar ozone loss by changes in temperature within the vortex that directly influence the formation of PSCs. Figure 11 represents the interannual evolution of sunlit VPSC above the Antarctic

and Arctic regions (top and bottom panels respectively). Larger sunlit VPSC values are expected in the SH than the NH due to much lower polar temperatures. Low values of sunlit VPSC are found for the years of low ozone loss and inversely as expected (see Fig. 9). Record of values sunlit VPSC are observed in 2020 for both hemispheres. As a consequence, very high ozone loss was found in the NH, and large but not record ozone loss in the SH. A linear trend was computed for VPSC from 2000, yielding an insignificant value in the SH and a positive value in the NH but significant only at $1\sigma$ level.

[Figure]

**Figure 11. Interannual evolution of sunlit volume of polar stratospheric clouds (VPSC) in the SH (top panel) and NH (bottom panel). The estimated robust trend (thick black line) and uncertainty level values of ±1σ (dashed black lines) since 2000 are added for both regions.**

Figure 12 presents the ozone loss value as a function of sunlit VPSC for each winter of the NH (triangles) and SH (inverse triangles). The figure highlights the difference between both hemispheres with much higher sunlit VPSC in the SH and consequently higher ozone loss. The range of sunlit VPSC in the SH varies between $2 \times 10^9$ and $5 \times 10^9$ km$^3$, which corresponds to an ozone loss between 36 and 55%. The range of sunlit VPSC in the NH is much smaller ($< 10^9$ km$^3$) but the dynamical range of ozone loss is slightly higher (4-27%). The figure highlights a quasi-linear relationship between ozone loss and VPSC in the NH (lower-left region in Fig. 12) and a different behaviour for larger ozone loss values due to the saturation of ozone loss in the lower stratosphere in the SH (e.g., Yang et al., 2008).

[Figure]

**Figure 11. Maximum ozone loss as a function of sunlit VPSC for each winter for the Northern and Southern hemispheres. The 68% inter-percentile range of ozone losses are also represented (see Sect. 3, Methodology). The colour code represents the years.**

In order to remove the influence of temperature interannual variability in the estimation of trends since 2000, a multi-parameter model was applied to the ozone loss dataset of each region as presented in Eq. 1:

$$MOLoss(t) = SunlitVPSC\_contr(t) + t1 * (year(t) – 2000) + \in (t) \qquad (1)$$

where t is year since 2000, t1 is the time linear trend since 2000, $\in (t)$ is the ozone loss residual and SunlitVPSC_contr corresponds to the contribution of sunlit VPSC considering a linear fit for the NH and a parabolic fit for the SH due to the saturation of ozone loss in the lower stratosphere (Eqs 2 and 3, respectively)

$$Sunlit\_VPSC\_contrNH(t) = K_{0\_NH} + K_{1\_NH}*SunlitVPSC\_NH(t) \qquad (2)$$

$$Sunlit\_VPSC\_contrSH(t) = K_{0\_SH} + K_{1\_SH}*SunlitVPSC\_SH(t) + K_{2\_SH}*SunlitVPSC\_SH(t)^2 \qquad (3)$$

The regression coefficients in Eq. 2 and 3 are significant at 2σ level. The autocorrelation of residuals of ozone loss in Eq. 1 is weak and lower than 0.2, and the determination coefficient ($R^2$) is of 0.83 for the SH and 0.82 for the NH. Figure 13 (left panels) shows a good agreement between MOloss dataset (colour lines) and the regression model results (black lines) considering estimated sunlit VPSC contribution (black dashed line) and trend.

The difference between the maximum ozone loss and the regressed sunlit VPSC contribution (ROLoss) is calculated for each year of the corresponding hemisphere as follows:

$$ROLoss(t)=MOLoss(t)- SunlitVPSC\_contr(t)= t1 * (year(t) – 2000) + \in (t) \qquad (4)$$

Figure 13 (right panels) shows the ROLoss dataset for the SH (top panel) and NH (bottom panel), respectively. The residuals vary between approximately 0 and -8% for the SH and within ±5 % for the NH. A decrease is observed since 2000 in both hemispheres with a higher interannual variability in the NH. The linear trend estimated by the multi-parameter regression model in both hemispheres (Eq.1) is around 2 % dec[-1] and significant at 2σ. Unlike the other two metrics, this metric provides a potential detection of a negative trend in the NH at the limit of significance.

The multi-parameter model was also applied to ozone loss using only SLIMCAT simulations (not shown). All regression coefficients are significant at 2σ, except the quadratic regression coefficient in the case of the SH. A larger recovery rate is found with the model simulation in the SH with a negative trend of -2.8 ±0.8 % dec[-1] (1σ).

For the NH, a slightly weaker trend was found compared to the observations with a value of -1.4 ±0.7 % dec[-1], also with limited significance at 2σ.

[Figure]

**Figure 13. Left panels: Interannual evolution of the maximum ozone loss (colour lines) since 2000 for both hemispheres and regression model (black lines). Sunlit VPSC contribution (see Eq. 2 for NH and 3 for SH) is superimposed by dashed lines. Right panels: Interannual evolution of Residuals of Ozone Loss with respect to regressed ozone loss values computed following Eq. 1 to 4 for the SH (top panel) and NH (bottom panel). The estimated trend (thick black line) and uncertainty level values of ±1σ (dashed black lines) since 2000 are also represented for both hemispheres."**

Lines 465, acknowledgements need to be made for the NDACC data

"The data used in this publication were obtained from "NDACC PI name" as part of the Network for the Detection of Atmospheric Composition Change (NDACC) and are available through the NDACC website www.ndacc.org."

The NDACC webpage was already mentioned in Section Data Availability. The sentence was added in the following way:

"The authors thank **the** technical teams operating SAOZ instruments **and NDACC PIs for the consolidated data**"

Line 449. Please provide the link to the ERA5 data.

The following link was added

https://cds.climate.copernicus.eu/cdsapp#!/dataset/reanalysis-era5-pressure-levels?tab=form

---

## Author Comment (AC2)

**Reply to Anonymous Referee #2 review of manuscript acp-2023-788**

**Trends in polar ozone loss since 1989: First signs of recovery in Arctic ozone column**

**Andrea Pazmino on behalf of all co-authors**

We thank Anonymous Referee #2 for the time devoted to evaluate our work. We are particularly grateful for the warning on the wording using to explain the results related to metric 3 that we have taken particular care of. Your comments have helped us to improve our manuscript. Please find our answers below (in red)

**\*\*\* More specifics regarding the more major issues/comments**

- L34 and related discussion in the manuscript: I would argue that a 1 sigma "detection" is not really a detection with enough significance; it is a likely detection as opposed to a robust detection (at 2 sigma or more), and some scientists in various disciplines would argue for even stronger significance levels, in addition to the fact that one often cannot or does not include all possible error bars in the analyses. In this case, the assumption of a linear relation between ozone loss amounts and VPSC is just that, an assumption taken as "truth" and any departures from this "truth" signify something related to ozone trends or recovery. In my view (and hopefully in the views of others in this field), this is just an indirect method at suggesting there may be "signs of recovery" (per the title of your manuscript), which is a better language than a "first quantitative detection". I understand that there is often a desire to show a "first" in research, but this can be overdone, and science progress is usually obtained via multiple analyses over time, especially for inferring trends, and the Arctic can change enough that adding or subtracting years can make substantial differences in the results. Rather than using a bold assessment like "first quantitative detection", I would urge the authors to use a more cautious statement. Error bars here are a lower limit, especially since the same sort of analysis for the Antarctic region yields error bars that are significantly smaller than other metrics results, so this is somewhat suspicious to me just on this basis, in addition to the fact that this method is more indirect than the other two metrics. Please counter this argument if you feel that there is a strong reason to declare victory on the Arctic recovery signal based on just one indirect metric and at the 1 sigma level (at best). I am not convinced, at this stage, and I feel that more metrics and years are needed for such a bold statement (including a broader community assessment, such as another WMO report, for example). I would not try to argue the validity of line 33 too strongly, as long as line 34 gets deleted, or replaced by something like "We argue that this points to the first signs of ozone recovery in the Arctic springtime lower stratosphere." [Although I personally would probably say this "may point", being a cautious person on such matters.] Alternatively, please make the case regarding such a bold statement by performing more error analyses - but this will typically increase the error bars, so the case will just become even weaker, I predict. Moreover, given that the ozone recovery path depends on both ODS and greenhouse gas effects, it is also difficult to provide a robust attribution of slightly positive trends to one effect or the other, without more detailed analyses; there are not enough model results to compare to, in terms of what a model would predict for one effect versus a combination of effects, in general, with comparisons to any of the observations shown in this work. I am therefore going to remain skeptical of broad sweeping conclusions for the Arctic, especially (although some caution is also recommended for Antarctic ozone studies). In fact, your own words at the end of the manuscript show more restraint and caution (with a pointer to another reference as well), so I imagine you actually agree with my words of caution. I think this shows nice results, whether one wishes to claim a "first" or not, and this is what should be the more important conclusion, a good set of analyses with hopefully reasonable error bars, and without overstating the possible conclusions.

As you have mentioned, the metric 3 uses an indirect but well-known assumption of the relationship between ozone loss and Polar Stratospheric Clouds due to the essential role of the latter in the heterogeneous chemical processes involved in the ozone depletion. We understand your concern about this indirect method and we took your warnings on wording into account. For example, the title has been changes as follows:

Trends in polar ozone loss since 1989: Potential sign of recovery in Arctic ozone column

To perform this study more robustly, we have decided to apply a multi-parameter regression model on the ozone loss dataset since 2000 using the Sunlit VPSC and a linear trend as proxies as shown in Equation 1:

$$MOLoss(t) = SunlitVPSC\_contr(t) + t1 * (year(t) - 2000) + \in (t) \tag{1}$$

where t is year since 2000, SunlitVPSC_contr(t) is the term corresponding to the linear (HN) or parabolic (HS) contribution of Sunlit VPSC, the variable t1 is the regression coefficient of the time proxy and $\in (t)$ is the fit residuals. The contribution of Sunlit VPSC is represented by Eq. 2 for the NH and Eq. 3 for the SH:

$$SunlitVPSC\_contrNH(t) = K_{0\_NH} + K_{1\_NH}*SunlitVPSC\_NH(t) \tag{2}$$

$$SunlitVPSC\_contrSH(t) = K_{0\_SH} + K_{1\_SH}*SunlitVPSC\_SH(t) + K_{2\_SH}*SunlitVPSC\_SH(t)^2 \tag{3}$$

where *K* are the regression coefficients of the respective proxies mentioned above.

All the regressions coefficients are significant at more than two standard deviations. The linear negative trends of ozone loss are significant and similar for both hemispheres, presenting slightly larger values than the previous results: $2.00\pm0.97$ % dec$^{-1}$ for the NH and $2.21\pm0.67$ % dec$^{-1}$ for the SH.

Those results confirm our previous ones but, as you mentioned, it is necessary to consider with caution the significance of the trends considering the error bars. The lines 33-34 were therefore changed as follows:

"Metric 3 provides a negative trend in Arctic ozone loss residuals with respect to the VPSC fit of **-2.00 ±0.97 (1σ) % dec$^{-1}$**, with **limited** significance at **2σ** level. **With such metric a potential** quantitative detection of ozone recovery in the Arctic springtime lower stratosphere **can be made**."

The Section 5.3 was changed introducing this new method and taking also into account the concerns of Referee 1. Please see the new Section 5.3 in the answer to Referee 1.

- As a related comment regarding Fig. 11, if one wants to claim enough robustness in the result and error bars, one should try to use 2 or 3 years less (or more) at the beginning or end of the series, to see how this affects the results and error bars. I think it is best, again, to be cautious in terms of a "robust detection" comment, unless the analysis is at least significant at the 2 sigma level, with enough sensitivity analyses as well.

The sub-section 5.3 was changed as explained above using a more robust method by applying a multi-parameter regression model. The low residuals values and the good correlation between observations and model (R~0.92) gives us confidence in the results.

- In addition, why not show what a polar-focused model would predict for such a metric, if one could add some credibility to the conclusions (in the Arctic especially) in terms of consistency with expectations.

Thank you for this recommendation and we have mentioned the results based on ozone loss from the SLIMCAT model in the last paragraph of the updated sub-section 5.3

- Also, of some interest, is there not a model-derived ozone loss onset date curve that can be compared to the Figure 9 results? What does this (or would this) show? Would this not be a useful comparison for what might be expected? This is not directly tied to ozone loss (but I do understand its connection to this). Any comments about this (in the text or at least as part of a reply) would be appreciated, since this might be worth considering as an added comparison (although not necessarily in this particular work).

This metric is only sensitive to large increases of ozone loss during the winter. This is then appropriate for the SH. The following Figure AR2.1 presents the onset days estimated from ozone loss values obtained from SLIMCAT simulations. The trend estimations were performed before and after 2000. All trends estimated by independently robust linear regression are significant at least 2σ. As for onset datasets from observations ozone loss, the lower trend values are observed for the threshold of 20% and the highest ones for 40% of ozone loss before and after 2000. The positive trends are slightly higher and vary between $4.4\pm1.0$ and $6.1\pm1.8$ day dec$^{-1}$. The ratio between the trends before and after 2000 of each ozone loss onset dataset is higher compared to the ones obtained from observations. It varies from -0.5 to -0.3. As expected, the model onset dataset presents a faster recovery than the observations (-0.3 to -0.2).

[Figure]

**Figure AR2.1: Onset day when 10-day averaged ozone loss reach a particular ozone loss value: 20, 25, 30, 35 and 40% for the SH using SLIMCAT simulations. Robust linear fits before and after 2000 are also shown (dashed lines).**

The following sentence was added in L362:

"The onset dataset obtained from SLIMCAT model simulations exhibits larger trends since 2000 that are significant at $2\sigma$ (not shown). The trends vary from $4.4\pm1.0$ to $6.1\pm1.8$ day dec$^{-1}$. The ratio between the trends before and after 2000 of each ozone loss onset dataset vary from -0.5 to -0.3 showing a faster recovery considering SLIMCAT simulations than using the SAOZ-MSR2 merged dataset, as already found using the ozone loss metric 1 (see Sect. 5.1)."

**\*\*\* More minor corrections and suggestions**

- L22: Add "column" between "cumulative" and "loss" for clarity.

Done

- L23, Abstract: since there are somewhat complicated calculations that involve more than just linear trends, as stated on line 27 (parabolic), this seems somewhat simplified of a description, even if the Abstract has to be simplified and short.

Maybe consider the following wording: "Three metrics are used in trend analyses that aim to assess the ozone recovery rate over both polar regions: ..."

The sentence was modified as suggested. Thank you.

- L28: I think you mean (or should use) standard deviation as part of the error analysis (see other comments above), instead of standard error, or justify the use of this terminology better.

Thank you for this remark. Everything has been adjusted using standard deviation.

- L29: you should be less vague and specify what threshold refers to here, what quantity (ozone column), instead of making the reader guess (if/as the manuscript has not been fully read at this point).

The threshold values were added in the abstract as follows (L29)

"For metric 2, various thresholds were considered **at the total ozone loss values of 20, 25, 30, 35 and 40%**, …"

- L29: I would suggest "all of them showing a time delay as a function of year, in terms of when the threshold is reached."

Adopted. Thank you.

- L32/33: "the difficulty in finding a threshold value in enough of the winters."

Adopted. Thank you.

- L63: wildfires events --> wildfire events

Done

- L72: change "concentrations/columns(?)" to just "columns".

Done. Thank you.

- L120: SAOZ ozone data could be more specific SAOZ total column ozone data?

OK

- L133: delete the period after "used".

Done. Thank you

- L173: delete parenthesis after "merged dataset"; also change "bias" to "biases".

Done. Thank you to highlight all these errors.

- L176, Figure 1 caption: please specify the year in the caption also (position of the 2021/2022 vortex edge...").

Done

- Figure 4: Please try to plot the thin black lines last, so they can be better seen on both panels. Consider making them slightly thicker as well. Should the y-axis title have a space between O3 and loss (maybe not, if you are referring to a specific variable instead of actual words). Also change "Day of the Year" in both panels (x-axis label) to "Day of Year" (as used in Figs. 5,6,7).

Thank you for your detailed work reviewing this paper. It is much appreciated.

You will find the new Figure 4 here below, following your recommendations.

[Figure]

- L235: For Figure 4, please specify in the text at which day of year the maximum amplitude of ozone loss between recent winters occurs (for both hemispheres). This will help the reader.

Thank you for your suggestion. The information was added in the text (L234)

"The maximum ozone loss is reached between mid-January and the end of March for the NH and between the end of September and mid-October for the SH."

- Figure 7: vortex edge as defined in Pazmino, but does that follow from Nash et al. (could specify here as well, if so, in addition to the text). Also correct the typo in NH y-axis (Gradiant --> Gradient).

Indeed, the definition of the vortex was done using Nash et al. but here we prefer to specify the definition for the Gradient, as done in Pazmino et al., 2018 which is based partly on the Nash et al. definition.

- Figs. 5,6,7: please change the thin black lines so they are plotted on top, for better visibility (and consider making them slightly thicker as well).

The figures were updated as Fig. 4.

- L253: and so was the vortex stability (rather than "and as well as the vortex stability").

Adapted

- L254: August, linked to a wavenumber 1 event,...

Done

- L255: upper levels, with an associated decrease in size.

Done

- L262: heat flux increases rapidly at the end

Done

- L264, slowing down rapidly thereafter.

Done

- L268: event, of a magnitude similar to the anomalies related to the Calbuco...

Done

- L271: mean T anomaly of -10.1 +/- 4.5 K, as in 2018, but with a much larger sunlit VPSC than in 2018.

Done

- L272: The persistently low temperatures [or The persistently cold lower stratosphere...]

Thank you for the suggestion. The sentence was changed by "The persistently cold lower stratosphere..."

- L273: led to an acceleration of the October ozone loss...

Done

- L276: (Fig. 6), and the strength...

Done

- L280: The sunlit VPSC values are similar...

Done

- L283: Fig. 4), which lies within the climatology.

Done

- L293: Do you mean "The strength of the vortex edge exhibited values larger than climatology...?

Yes exactly. We have changed as suggested:

"The strength of the vortex edge exhibited values larger than climatology..”

- L294: vortex led to moderate ozone loss; also please specify again (if need be) where the ozone loss error bar values come from (or refer back to that discussion)., and if these represent one standard deviation (presumably not two).

The 1σ was added at the first time the ozone loss was mentioned in subsection 4.1 (L250) et 4.2 (L294)

- L300: ozone loss of the 2019 warm winter

Done

- L304: temperature anomalies at the 475 K level... and the mean anomaly value for the whole winter reached -5.3 K, as in 2018.

Done

- L306.307: final warming mode, also shown by the low values...

Done

- L308: persistent low temperatures less than ...

Done

- L319: a possible recovery path of total ozone... [or recovery rate]

The sentence was changed by “… a possible recovery **rate** of total ozone columns …”

- Figure 8: please make the grey legend stand out more; for example, use larger fonts for the legends and say % dec-1 to shorten the units and legend length, and use a bolder font if needed.

The figure was updated as suggested.

- L344: are positive (1.0 +/- 2.2 %dec-1)

Done

- L348: values, as we might expect a later onset... in relation to lower...

Done

- L350: what is the ozone loss time dataset? Is this not the same as the ozone loss onset days "are used instead of total ozone columns"... (?)

The sentence was changed as follows:

"In this study, the ozone loss **onset days** dataset is used instead of total ozone columns **onset days dataset** in order to consider chemical processes only."

- L371: You say a "3rd order polynomial..." and also mention a parabolic fit; to me, a parabolic fit is a 2nd order polynomial (i.e., a quadratic). Please clarify.

The subsection 5.3 was changed. Please see previous answers.

- L369-L371, I would say "dynamic range" or just "range" really; dynamical could appear to refer to something atmospheric...

Thank you for highlighting that. The word "dynamical" was removed.

- Figure 10: It might be interesting to try a linear fit after 2000 for the SH; not necessary for this paper, just a thought (how would that affect the results?).

A linear proxy was considered to represent the sunlit VPSC contribution on Ozone Loss in the SH as adopted in Sect. 5.3 for the NH. A larger negative trend of -2.8 ±0.8 % dec$^{-1}$ was found compared to -2.2 ±0.7 % dec$^{-1}$ using the quadratic relationship. Those values are within the standard deviation. Considering a linear relationship between O3 Loss and sunlit VPSC, the determination coefficient is weaker but still large with a value of 0.78 instead of 0.83.

- L419: was calculated, but it is not significant.

Done

- L420-422: It would flow better if the sentence "Regarding the SH..." was placed one sentence above, directly after the SH comment. Then one could just say "This metric appear sensitive..."

Done

- L433: Note that this trend is similar to the SH trend.

Done

- L436: add a comma after "datasets".

Done

---

## Author Response (AR2)

I would like to thank the anonymous reviewer for his remarks on the revised version of the manuscript acp-2023-788.

**Very minor (technical) suggestions for the revised version.**
- the overpass criteria were going to be clarified for the MSR2 values selection; please ensure that the simple sentence that was supposed to be included actually gets included (it did not seem to be - at line 128, presumably).

The sentence was completed as mentioned in the previous answer to referee 1.

- Abstract, line 33, please change "limited significant" to "limited significance"; also the last sentence in the Abstract should read "With such a metric, a potential quantitative detection...can be made."

Done

- Line 161 should read "The number of observations in the Arctic vortex ..."

OK

- Figure 11 on page 18 should be numbered Figure 12.

The number of the figure was changed. Thanks